# Understanding the Technical-Scientific Gaps of Underutilized Tropical Species: The Case of *Bactris gasipaes* Kunth

**DOI:** 10.3390/plants12020337

**Published:** 2023-01-11

**Authors:** Yasmin Verçosa Kramer, Charles Roland Clement, Josiane Celerino de Carvalho, Andreia Varmes Fernandes, Carlos Vinicius Azevedo da Silva, Hector Henrique Ferreira Koolen, Jaime Paiva Lopes Aguiar, Adriano Nunes-Nesi, Marcio Viana Ramos, Wagner L. Araújo, José Francisco de Carvalho Gonçalves

**Affiliations:** 1Plant Physiology Graduate Program, Departamento de Biologia Vegetal, Universidade Federal de Viçosa (UFV), Avenida PH Rolfs, s/n, Viçosa 36570-900, Brazil; 2Department of Technology and Innovation, National Institute for Amazonian Research (MCTI-INPA), Avenida André Araújo, 2936, Aleixo, Manaus 69011-970, Brazil; 3Laboratory of Plant Physiology and Biochemistry, National Institute for Amazonian Research (MCTI-INPA), André Araújo Avenue, 2936, Aleixo, Manaus 69011-970, Brazil; 4Grupo de Pesquisas em Metabolômica e Espectrometria de Massas, Escola Superior de Ciências da Saúde, Universidade do Estado do Amazonas, Avenida Carvalho Leal, 1777, Cachoeirinha, Manaus 69065-000, Brazil; 5Coordination Society of Environment and Health and Laboratory of Physical Chemistry of Food, National Institute for Amazonian Research (MCTI-INPA), Avenida André Araújo, 2936, Aleixo, Manaus 69011-970, Brazil; 6Departamento de Bioquímica E Biologia Molecular, Centro de Ciências, Universidade Federal do Ceará (UFC), Campus do Pici, Benfica, Fortaleza 60020-181, Brazil

**Keywords:** agronomic traits, biotechnological applications, chemical composition, crop fruit, genetic resources, germplasm banks, peach palm plantation

## Abstract

The extraction and commercialization of palm hearts is the most profitable activity involving the peach palm *(Bactris gasipaes*), while consumption of its fruits is limited to Amazonian communities. The excessive attention paid to the implementation of germplasm banks contributed to the lack of development of high-performance varieties, limiting the production and consumption of peach palm fruits and by-products. In addition, with the fragmentation of the Amazonian rainforest, wild populations are in danger of extinction. The species domestication, initiated by Native Amazonians, generated a large variety of peach palm populations, as evidenced by the diversity in fruit sizes and quality. Some advances in agronomic traits also took place. However, more research needs to be conducted to understand the implications of climatic changes on plant physiological performance. Indeed, the key point is that the exploitation of the full potential of *B. gasipaes* has not been completely exploited. Therefore, understanding the state-of-the-art research on the peach palm with a focus on its underutilized resources is essential for expanding plantations and, consequently, promoting the market expansion of the peach palm as a fruit crop.

## 1. Introduction

*Pupunha* (Brazil), *pejibaye* (Costa Rica and Nicaragua), *pijuayo* (Peru), *perapon* (Guyana), *chontaduro* (Colombia and Ecuador), and peach palm (in English-speaking countries) are some of the common names of *Bactris gasipaes* Kunth, a caespitose palm considered to be “multi-purpose”. This is the reason for its great importance to Native Amazonian populations, who started its domestication process [1,2]. The species is naturally distributed between Bolivia, Brazil (North and West-Central), Colombia, Costa Rica, Ecuador, French Guiana, Honduras, Nicaragua, Panama, Peru, and Venezuela [1,3]. In addition, the peach palm has also been cultivated in Hawaii, Indonesia, Malaysia, and Réunion Island [4].

Modern populations of cultivated peach palms have high heterozygosity, resulting in a vast variety of fruit morphology and, therefore, in fruit quality. This phenotypic richness could contribute to obtaining genetically improved individuals, but it also implies obtaining various by-products with different qualities. However, even with its multiple potentials, the peach palm has been considered an economically underutilized species [1].

The excessive focus on market prospection directed R&D efforts to the implementation of germplasm banks, the prospection of an ideotype for palm heart production, and failed attempts at creating new niches in the market for fruit by-products [1]. High maintenance costs, susceptibility to diseases caused by high-density plantations, and other problems of an infrastructural and political nature led most of the projects involving germplasm banks to fail [1,5]. The lack of success in developing high-quality varieties restricts the expansion of peach palm production. Moreover, the seed selection of limited matrices contributes to genetic erosion [1,6]. In addition, the increasing fragmentation of the Amazon rainforest due to deforestation, illegal fires, livestock production, and agro-industrial expansion affect the wild populations of *B. gasipaes*, which also leads to genetic erosion. The loss of genetic variability may affect potential genetic breeding programs [6,7]. In contrast, recent advances in the establishment of in vitro culture protocols for the species may contribute to both the conservation of threatened genotypes and advances in the genetic transformation of the species.

To promote peach palm as a fruit crop, it is important to understand the current state of the species. Therefore, this review focuses on updating technical knowledge, existing limitations, agronomic aspects, and advances in the field of biotechnology with peach palms. We first describe the general aspects of peach palm biology, its regional use, and biotechnological applications, followed by an overview of agronomic practices and genetic resources of this species. Finally, we discuss how the current knowledge may be integrated, with the aim of identifying profitable activities involved in the peach palm market and discovering novel products with biotechnological applications.

## 2. Peach Palm Is a Domesticated Palm in Neotropics

Naturally distributed throughout northern South America and southern Central America, the peach palm is considered the only domesticated palm in the Neotropics [8,9]. The domestication of *Bactris gasipaes* was guided by centuries of cultivation by Native Americans due to its multiple uses. In addition to the fruits that were later diversified and adapted to each population’s traditions, wood was also used for manufacturing tools or as construction material [8,10].

Until the end of the 20th century, the peach palm was considered a cultigen [11]. Considering the cultivated populations, Mora-Urpí and Clement [12] proposed a classification based on fruit morphometric characteristics. The populations were classified into three groups: microcarpa, for fruits with a weight below 20 g; mesocarpa, for fruits between 20 and 70 g; and macrocarpa, for fruits with a weight above 70 g. Considering the suggestions of Mora-Urpí and Clement [12], as well as of Harlan and De Wet’s [13] gene pool approach, Henderson [3] proposed a revision of Bactris based on morphology, cytotaxonomy, and palynology. The peach palm was recognized as *Bactris gasipaes* with two varieties: var. *chichagui* (types one, two, and three) and var. *gasipaes*. The var. *chichagui* comprised wild populations with small and oilier fruits, and the var. *gasipaes* comprised domesticated populations with large and starchier fruits.

Molecular analyses involving peach palms began at the end of the 1990s. Clement et al. [14] used allozymes to investigate the relationship between three peach palm populations from San Carlos (Costa Rica), Yurimaguas (Peru), and Benjamin Constant (Brazil). High intraspecific heterozygosity was found, a result that was reinforced later with the use of RAPD markers [15], demonstrating the proximity between Benjamin Constant and Yurimaguas, Iquitos and Pará, and finally, San Carlos.

Rodrigues et al. [16] also investigated the genetic variability of seven morphologically distinct populations using RAPD markers. Contrasting results were found compared to those presented by Clement et al. [14] regarding heterozygosity between Benjamin Constant (Putumayo landrace), Yurimaguas (Pampa Hermosa landrace), and San Carlos (Guatuso landrace). In the same study, the formation of distinct evolutionary groups was visualized, reinforced posteriorly by Araújo et al. [17]. The group referring to Central America would be composed of representatives of Utilis (ex-Tuíra) and Guatuso landraces. The Western Amazon group would be composed of populations from Putumayo (former Solimões, including Benjamin Constant) and Pampa Hermosa landraces. Finally, the group referring to the Southeastern Amazon would be composed of Pará landrace and Acre populations (var. *chichagui*). These results support the hypothesis of landraces (locally improved, domesticated cultivars), which had been pointed out, until then, by studies based on the morphometric differences between populations [12,16]. In addition, it is argued that the morphometric and phylogenetic relationship of the landraces indicates at least two migratory routes for peach palms: one through the Eastern Amazon and the other through the Northwest, corroborating the idea of multiple domestication events.

This hypothesis was contradicted by other studies using cpDNA and nDNA analyses through the recognition of an incipiently domesticated genotype: the var. *chichagui* type three [18,19]. Hernández-Ugalde et al. [20] supported the idea of at least three domestication sites, providing a basis for what was pointed out previously [16]. Galluzzi et al. [19] and Clement et al. [18] defended the idea of multiple domestication events, based on two major distributions: across southwest Amazonia to the west (and beyond) and the second across the Madeira River region, to eastern Amazonia (Figure 1). It was also argued that the distribution of peach palms was not only correlated with indigenous management but also with the geological events that occurred in the South American continent between the Miocene and the Pliocene [18,19,20]. In addition, it was hypothesized that there were wild and cultivated populations that had not yet been recognized [20].

A recent complete plastome characterization of var. *gasipaes* and var. *chichagui* demonstrated that the differences between these two varieties are also visualized at the molecular level [21,22]. With the advances of molecular tools and new findings regarding the evolutionary processes, further studies are still required to better understand the infra-specific relationships among the landraces, as well as the revision of *B. gasipaes* nomenclature [18].

### 2.1. General Morphology of Peach Palm

The peach palm has a slender trunk that can reach up to 25 m in height [3,23]. The trunk sustains a crown with up to 20 pinnate leaves that may vary from two to four meters in length. The monoecious inflorescences in the form of racemes occur in the axils of senescent leaves and comport small yellowish flowers. The staminate flower is deciduous, diclamide, and has six stamens. The pistillate flowers have an annular calyx and tubular corolla, and the presence of staminoids with a protogynous development (Figure 2) [3].

After maturation, the inflorescence becomes a bunch containing about 100 fruits, rarely up to 1000. The fruit can be starchy or oily and vary between 10 and 250 g in weight for var. *gasipaes* and between 0.5 and 5 g in weight for var. *chichagui* [3]. The fruits have either a rounded or ovoid form, and the fibrous epicarp may vary from a yellow ocher to a reddish color [3,8]. A natural segregant that produces white fruit is now also documented [25].

The seed has a lignified endocarp with a germination pore, an oily-fibrous, and homogeneous endosperm with a light color (almost white), a cotyledon, and a peripheric conic embryo [24,26]. Germination occurs between 38 and 133 days or up to 150 or more in non-ideal conditions [24,27]. The long time of germination is related to physiological dormancy, but this characteristic also presents a great variation in duration, not being well understood [28]. The germination is classified as cryptocotyledonar-hypogynous [24].

The peach palm can sprout shoots at the trunk base, characterizing it as a caespitose. According to Edelman and Richards [29], there are five types of vegetative branching in Arecaceae, with lateral axillary branching similar to that observed in the peach palm. The roots rise from the base of the trunk, creating a fasciculate system that penetrates two meters into the ground, and is distributed five meters around the plant, each containing absorbent hairs. It also forms adventitious roots on the base, forming a clump where the shoots aggregate, which may represent an adaptation to flooded environments [24,30,31].

Most peach palm genotypes present dark or yellowish-brown spines on the trunk internode, on the petiole and the leaves’ abaxial side, on inflorescence peduncles, and also on the shoots [3,8]. The occurrence of spines is a form of mechanical protection against plagues and herbivory but is also related to water caption and absorption [32]. However, the importance of spines for fruit and palm heart production was reviewed, and no direct correlation between the spines and higher yield was found [33].

### 2.2. Ecological Relationships

The peach palm is widely distributed in the Amazon Basin and has a hole in forest systems. A great diversity of mammals and birds (e.g., deers, agoutis, *Penelope obscura* or *jacuaçu*, toucans) consume the fruits in the bunch or that fall under the tree. These animals are also extremely important in aiding the seed dispersal of wild populations [23]. Trichomes are also consumed by some insects. The *B. gasipaes* trichomes are modified and present brachysclereids (sclerenchyma) [34]. They have a high percentage of lignin and some proteins, glucose, saccharose, palmitic, oleic, and linoleic fatty acids. This structure is consumed before pollen by Diptera, Coleoptera (mostly Curculionidae), Hymenoptera, Hemiptera, Neuroptera, Lepidoptera, Orthoptera, and Dictyoptera, and has a similar function to gastroliths [34,35].

It has been demonstrated that the peach palm can be colonized by different morphotypes of the *Acaulospora*, *Glomus,* and *Scutellospora* genera [36]. The arbuscular mycorrhizal fungi (AMF) develop internal and external mycelium with arbuscules on the root cortex and can also present some vesicles that are external to the roots [37]. They produce a glycoprotein known as glomalin that involves the glomospore and hyphae. This aids plant carbon absorption/fixation and contributes to natural soil fertilization and colloid formation, helping to provide resistance against pathogens and abiotic stress [36,37,38]. Besides the permanence of AMF on peach palm roots all year [36], the symbiotic relationship between AMF and *B. gasipaes* differs among the progenies, as observed by Clement and Habte [39]. In progenies of Pampa Hermosa, Putumayo, and Guatuso landraces, differences were observed in their dry matter accumulation when associated with AMF, e.g., 17%, 54%, and 64%, respectively. No correlation among more primitive landraces or dependency on AMF was observed, although there may be different physiological requirements among these landraces.

Chalita et al. [40] identified *Pseudomonas putida*, *Burkholderia ambifaria,* and *Burkholderia* sp. in agroforestry systems containing peach palms. The *Pseudomonas* genus is widely reported to be a solubilizing bacteria and is related to auxin (AIA) and the precursor of ethylene synthase (ACC deaminase) production. The *Burkholderia* genus is recognized for fixing N_2_ and can be associated with mycorrhizal fungi, as well as having the potential to biologically combat the oomycete *Phytophthora palmivora*. Moreover, *Burkholderia* is considered to be a plant growth-promoting rhizobacteria [41]. In addition, the aforementioned genera are considered biofilm formers, which have been of interest to the agroindustry due to their capacity to enhance soil fertility and crop production [42].

## 3. Agronomic and Physiological Aspects of Peach Palm

### 3.1. Practices in B. gasipaes Cultivation

The most effective way to propagate the peach palm is via seeds; however, this type of propagation has been one of the limiting factors for its agricultural expansion [43]. This is related to the lack of improved peach palm seed production and phytosanitary control, which can contribute to a reduction in the material quality, germination rates, and irregular yield [44,45].

After matrix selection, fruits should be harvested almost fully ripened to conserve the material against soil pathogens and increase germination rates. The pulp must be removed to prevent fermentation. After the manual removal of the tegument, the seeds must be soaked in water for one or two days, with a daily change of water. It is also recommended to undertake a cleaning step with commercial hypochlorite and water (100 mL NaClO to 1 L of water) for 15 min, followed by a wash in running water [46]. Fungicides can also be used for seed asepsis [43].

Peach palm seeds are recalcitrant and the embryo viability is compromised by water content loss, with no germination in water content below 17% [27]. High humidity associated with warmth can favor fungal contamination, therefore, sowing right after the cleaning step is recommended. The seeds can also be conserved for short periods with the manutention of water content (to maintain the embryo viability), temperature (between 15 and 18 °C), and packaging [43].

For sowing, nursery bags or beds are used with a substrate of light material (such as sand or sawdust). Mature manure can be incorporated into the substrate in a proportion of 1:1. The seeds must be sowed at a 2 cm depth in the substrate. The use of shade cloths is highly recommended to protect the seedling from direct sunlight. The germination occurs between 30 and 120 days in ideal conditions. Daily irrigation (or every 2 days depending on the type of soil and the season) is recommended [43,46].

Peach palm is known as a rustic plant; however, for better performance, soil preparation and fertilization are necessary. Before transplantation, ploughing and harrowing are essential for compacted soils and/or heavy structures (e.g., Ultisol). Fertilization for production is also commented on. For small-scale production, the use of plant residues (e.g., dead leaves and dropped fruits) or manure may be sufficient. For a larger scale, the use of chemical fertilizers (for N, P_2_O_2_, K_2_O, S e B, especially) must be initiated six months after transplant. Biofertilizer alternatives are currently available in Brazil’s market and may be a more sustainable way to increase production. In both scale categories, regular management of weeds is required [46,47].

The first fruit harvest starts in the third year after the transplant. The main difficulties related to bunch harvest are the height and presence of spines. When the plant has spines, the harvest can be performed with the aid of a mesh (to catch the bunch when it falls after cutting) and a long stick with a knife similar to a scythe (or a long scythe as well). The remotion of spines (or plants without spines) allows climbing and direct bunch harvest [48]. The harvesting of the palm hearts can be conducted when the first internode is visible (18th month) or when the trunk achieves 1.5 m [46,47].

Additionally, the use of shoots (asexual propagation) could help maintain the preferred characteristics of the parent plant i.e., fruit production, precocity, and absence of spines. Due to its caespitose characteristic, *B. gasipaes* can be propagated by separating the shoots from the mother plant. To propagate it asexually, it is more appropriate to use shoots that are between 30 and 60 cm. The presence of a primary root system could help with adaptation when removed from the clump. This method is divided into two phases, i.e., the field phase, consisting of the selection of the shoots and separation of the clump, and maintenance in the greenhouse for 30 days, and then transfer to nursery bags with the use of shade cloths in 50% of the sunlight [49]. However, this propagation method still has a lower success ratio when compared to the use of seeds. The actual protocols of peach palm macropropagation require methodological improvement, and the use of growth regulators must be better understood.

The peach palm is commonly associated with monocultural practices (for palm heart production), but due to forest multi-strata adaptation, it also has a good yield in agroforestry systems. *Theobroma cacao*, *Theobroma grandiflorum*, *Manihot* sp., and palms such as *Euterpe* spp. are some of the species that can be associated with peach palm [50,51]. The possibility of culture with other plants, such as *Coffea canephora* and *Coffea arabica*, peanut, rice, and sorghum, has also been studied. Some of these combinations with *B. gasipaes* showed an increase in physic-hydraulic quality, soil water content, and microporosity, which improves the product quality [52,53]. This activity is not only more profitable, but it can also be a way to recover degraded areas [54].

Peach palm culture has great potential for the expansion of fruit and palm heart production; however, efforts to better understand the management necessities are still needed. The selection of high-performance genotypes and the development of improved seeds through genetic transformation may be a way to maintain a large genetic base and improve fruit and palm heart yield. In the same way, asexual propagation is promising; however, it is necessary to investigate efficient ways of rooting, the physiological impacts of this type of propagation, and how much they interfere with productivity. In addition, agroforestry systems may represent a union between productivity and conservation. Although promising, it is still necessary to expand experiments with different crop combinations and compare the impacts on production and ecological services provided by these systems.

### 3.2. Abiotic Stress

As a tropical crop, the peach palm has a high-water demand. Water stress is more visible in subtropical cultures since there is expressive seasonal variation [55]. This type of stress induces stomatal closure and the dynamic photoinhibition of the secondary photosystem (PSII) as a protective mechanism against evapotranspiration water loss, indicating peach palm resistance to drought [55,56]. Aiming to investigate the response of peach palm plantlets to waterlogged soil, a study revealed that hypoxic conditions reduce N and K absorption, chlorophyll content, and stomatic conductance. In addition, they increase the content of soluble sugars in leaves and roots. Although the presence of the adventitious roots does not seem to promote long-term survival, it may help with increased tolerance to hypoxia conditions [30].

The major nutrient dependency is nitrogen, whose deficiency can cause chlorosis followed by necrosis in senescent leaf margins [57,58]. It is also associated with the low production of chlorophyll due to chloroplast modification and a reduction in the AMF symbiotic interaction [59]. The peach palm is also affected by potassium deficiency, which causes chlorosis and necrosis on the leaf blade. A lack of calcium causes the uneven development of leaves, associated with putrescine accumulation on the lesions. Other macronutrient deficiency symptoms can be listed, i.e., a reduction in plant growth (P), internerval chlorosis (Mg), or, on newer leaves, a substitution of green to a green-yellowish color (S) [57,58].

Micronutrient deficiency is also reported; Fe, B, and Zn omission promotes chlorotic leaves, the death of the terminal meristem and reduction in root proliferation, and narrow leaves with necrosis on the tip, respectively [57]. In addition, the omission of Na is related to chlorosis and necrosis in the leaf tips [58]. In another study, Fernandes et al. [60] analyzed the peach palm seedlings’ resistance to salinity. The best treatment was with Na at 1 mmol L^−1^ and Cl at 0.5 mmol L^−1^, which increased plant development. The worst treatment was with NaCl at 15 mmol L^−1^, indicating that peach palm is sensitive to saline soils (considering Abrol et al. [61] soil salinity classes).

Considered a rustic plant, the knowledge about peach palm performance in abiotic stresses must be expanded. In addition, it is necessary to investigate potential differences between genotype performance under abiotic stress, in which information could help the selection of stress-tolerant genotypes for seed improvement programs.

### 3.3. Pests and Diseases

The mite *Retracrus johnstoni* is known to promote chlorotic and necrotic spots which can affect peach palm productivity [62]. Peach palm is also affected by Coleoptera, e.g., *Rhynchophorus palmarum,* and *Dynamis borassi* [63,64]. In addition, *D. borassi* is a vector of the nematoid *Bursaphelenchus cocophilus*, which can represent a threat to coconut intercropped with peach palm [64]. The peach palm can also be attacked by caterpillars from Lepidoptera of the Noctuidae family [65]. The triatomine *Leptoglossus conchoides* is related to early fruit fall [66]. For these pests, better irrigation management to avoid the ideal conditions for pest establishment is appropriate, as well as the use of pesticides for species whose life cycle is known. In addition, the application of synthetic pheromones, such as rhynchophorol, is also recommended for the ethological control of *D. borassi* [67].

Fungal infection is the most common cause of a reduction in production and plant death in the peach palm. The seeds and fruits can be infected by common soil fungi, i.e., *Aspergillus* spp., *Trichoderma* spp., and *Fusarium* spp. reducing the germination rate [43]. Fruits are also affected by *Thielaviopsis* spp.: a promoter of black rot [68]. The infection by *Colletotrichum gloeosporioides* causes anthracnose in different parts of the plant. The leaves can be infected by *Bipolaris bicolor* and *Curvularia* spp., which cause leaf spot disease and leaf blight, respectively. *Cercospora* sp. and *Alternaria* sp. are also related to leaf spots. Moreover, *Fusarium* sp. is related to systemic yellowing, wilting, and death in plants [68,69,70]. In addition, infection by the bacteria *Pantoea stewartii*, associated with *Fusarium* sp., is also related to leaf necrosis. *M. hemipterus* and *R. palmarum* are reported as vectors of pathogenic bacteria [71,72].

There is a growing concern about the oomycete *Phytophthora palmivora*, which causes rot in the base of the stem, principally at low temperatures, to represents a notable risk to plantations in subtropical climates. This disease can cause the loss of more than 80% of the production of fruits and palm hearts. Its identification in the field is difficult, and there are no known ways to combat the disease [46,73].

In general, the management of humidity, water, and heat in order to reduce plantation density, along with the application of copper or boron solutions, are recommended for disease control. The use of insecticides or other biocides can be an alternative but may not assure the control of contamination and contribute to increasing microorganism resistance [70]. Good nutrition and association with symbiotic microorganisms could reinforce the peach palm’s resistance to these diseases and pests [37,38,40,70], however, more studies on the peach palm that focus on biological control are still needed.

## 4. Peach Palm Products: Diversity in Consumption, Chemical Composition and Biotechnological Application

Patiño [10] emphasizes that the use of peach palms by native Amerindians was encouraged by the properties of the wood and later by the fruit and other parts. The multipurpose uses of peach palms translate into the literal use of all parts of the plant [9]. For example, the dried leaves are used as the font of straw to manufacture materials such as baskets, mats, and body adornments; the palm heart is consumed in natura; the inflorescences, dried and macerated, can be used as condiment/flavoring; the wood is used to manufacture instruments, tools (such as knives, arrows, and bows), and as a construction material; and the roots are used as a large-spectrum vermicide, while other medicinal uses are also documented [2,10,74,75]. Besides its manufacturing application and use as a food source, the peach palm parts are also related to traditional folklore [2,10,75].

In Brazil, the peach palm is now mostly limited as a source of palm heart, being popularized mainly by the trade in brines and by the fruit that has its consumption limited to the regions where the species naturally occurs [1]. In this topic, some aspects of the peach palm market will be discussed, whose economic potential has not been fully exploited. Furthermore, the chemical composition of the fruits and their biotechnological applications will be described.

### 4.1. Palm Heart Is Present in International Market, but Fruits and By-Products Consumption Is Associated with Basal Market

In fact, the most profitable activity of peach palms is palm heart commercialization. The low activity of polyphenol oxidases and peroxidases confers the stability of the taste and color, allowing its commercialization in natura or in brines [47,76]. It is estimated that the international market for processed palm hearts (regardless of the species) is around US$ 500 million. Ecuador is the main exporter of canned palm hearts, totaling around US$ 65.6 million and 28.9 thousand tons in 2021, followed by Bolivia [77,78].

Over the last 15 years, Brazil has been losing space in the international market; in 2006, it was considered the third largest exporter, totaling around US$10 million and 1.7 thousand tons. However, in 2017 it was considered the eighth largest exporter, totaling US$1.47 million with exports of only 265.4 tons. Nowadays, Brazil still remains as the eighth largest exporter of processed palm hearts, totaling around US$1.77 million, exporting 284.6 tons in 2021 [79,80]. This has been justified by the lower quality of the processed palm heart, extractives’ practices, and high internal consumption [81].

In Brazil, palm heart comes from different species (*Bactris gasipaes*, *Euterpe edulis*, *Euterpe oleracea,* and the invasive palm *Archontophoenix cunninghaniana*) natural populations and exploitation of plantations [82]. According to the Brazilian Institute of Geography and Statistics (IBGE), in 2006, palm heart obtention was mostly carried out by extractive activity, amounting to about 53% of the total production [83,84]. In 2017, around 90.8% (approximately 93 thousand tons) came from plantations in the states of Santa Catarina, São Paulo, and Paraná [85,86]. However, the lack of objective information regarding the species used for palm heart production makes it difficult to understand the impact of peach palms on the national and international markets. Likewise, there is little information about the distinct types of commercialized palm hearts (in natura and canned) as well as national consumption patterns.

As commented previously, the fruit is limited to the domestic market. In 2006, gross production reached around 6.7 thousand tons, amounting to approximately US$3.13 million [83,84]. An increase in production between 2006 and 2017 was visualized, hovering around 10.3 thousand tons and adding up to US$4.12 million [83,84,85,86]. However, this scenario may not reflect the direct consumption of the population. It is estimated that almost 60% of fruit production is wasted and/or not consumed by the local population. This is associated with the fact that the fruit is a drupe, which hinders its storage and longevity/quality of the fruit. It is also estimated that the population prefers the fruit in natura rather than processed [1].

Besides direct consumption after cooking, fruit pulp can also be added to lunch dishes. In Peru and Bolivia, the use of starchier fruits is related to the creation of a fermented drink known as *chicha.* In Brazil, *caiçuma* fermented drink consumption is limited to ethnic communities [23,47]. The fruits can also be used as raw material for gluten-free flour production. Peach palm flour can be used in bakeries; however, human consumption is still not widespread. Indeed, the use of this by-product as a base for animal feed has also been studied, but its economic potential is still not well understood [1]. The peels also pursue potential as a font of fibers and metabolites, such as carotenoids [87]. The seeds also represent economic potential: an oily endosperm proves to be a rich source of vegetable oil. However, the removal of the endosperms can be laboring due to the hard and woody consistency of the seed’s exocarp [88,89]. In Brazil, the fruits and by-products did not achieve a place in supermarkets but instead were found in street markets though they are difficult to obtain in the offseason [1]. In addition, the use of other parts of peach palms is even more limited, and little is known about the current production demand and market for these by-products.

Clement et al. [1] discussed the creation of market niches for peach palm fruit by-products in Brazil; however, the engagement of R&D has failed in the popularization of these items in recent decades. Bearing in mind the potential of the fruit and by-products, as well as the expansion of palm heart cultivation, authors have, therefore, classified peach palms as an economically underutilized species [1,48,75].

From this perspective, an increasing interest in this species was observed in the last 10 years, resulting in a rise in the knowledge about its chemical properties and biotechnological applications. The technological investigation of this species may represent a change in perspective to the point where the species presents its economic potential.

### 4.2. Fruit Cheminal Composition

The peach palm fruit, in comparison to other Amazonian fruits, such as *buriti* (*Mauritia flexuosa*), can be considered a reliable source of energy (Table 1). Furthermore, peach palm fruits present relatively low quantities of macroelements, providing minor contributions to daily allowances [90]. A study using flame atomic absorption spectrometry [91] detected lower quantities of these minerals in peach palm fruits than reported in the literature [90,92,93], a factor that may be directly associated with the material quality of the investigated genotype. Regarding micronutrients, the presence of selenium and chromium adds to their nutritional value since these elements are associated with blood sugar regulation, cholesterol control, and the strengthening of the immune system [90,94].

The peach palm fruit is considered a good source of fiber as it is mainly composed of pectic polysaccharides that create a linear, highly methyl-esterified homogalacturonan structure with minor portions of xylogalacturonan and type I rhamnogalacturonan [103]. Amylose represents 18.2% of total carbohydrates [104]. Yuyama et al. [90] commented that the protein value of peach palm fruit is low due to the total protein content and also to the reduced amounts of essential amino acids. Although it is not a robust source of these nutrients, it is possible to state that its consumption can be supplemental to other protein sources.

In addition, fruits have a high content of unsaturated fatty acids, especially oleic acid (Table 1). The consumption of long-chain unsaturated fatty acids is essential for physiological processes, whether through energy generation or maintenance at the cellular level. Moreover, in balanced amounts, their consumption plays a significant role in reducing the risk of developing vascular and neurodegenerative diseases [105]. The seed also presents itself as a major source of oil. The composition of the peach palm seed endosperm contains high quantities of lauric and oleic acids (33.3 and 24.3%, respectively) and minor percentages of sterols (Figure 3) [88,89].

It should also be noted that the consumption of the fruit should be undertaken after cooking due to anti-nutritional factors [89,93]. Therefore, despite the low content of essential amino acids, peach palms can be considered a fruit of high nutritional value. This is mostly due to its energy supply, oil quality, essential and non-essential mineral content in the human diet, and high availability of vitamins, and thus its consumption is strongly encouraged.

#### Specialized Metabolites

The peach palm fruit can be considered a complex pool of bioactive compounds (Figure 3). Among the terpenoids that occur in the lipid fraction of peach palm fruit, carotenoids are found in greater abundance than xanthophylls. De Rosso and Mercadante [106] compared the carotenoid content of the peach palm mesocarp to other species and revealed the high availability of these compounds in peach palms, surpassing palm oil (*Elaeis guineensis*). However, it presents an inferior content of carotenoids when compared to *buriti* (*Mauritia flexuosa*) (Table 2).

The high quantity of carotenoids in the peach palm fruit adds to its nutritional value. The consumption of bioactive compounds, such as carotenoids, helps in the fight against degenerative diseases and in the maintenance of human health [112]. In this case, the presence of long-chain unsaturated fatty acids in the pulp also favors the delivery of lipophilic compounds to the body. Furthermore, the bioavailability of carotenoids in the peach palm does not appear to be affected by cooking; therefore, it is a safe and highly bioavailable source of these compounds [113]. Additionally, peels are considered the greatest font of fiber and carotenoids [87].

Chisté et al. [113] observed that orange fruits have a greater number of bioavailable carotenoids than yellow fruits. On the other hand, lutein was more available in the yellow variety, accounting for 14% of the total carotenoid content. These discrepancies may explain the variety of colors found in peach palm fruits. In addition, peach palm fruits are also considered a good source of sterols (Table 2), with β-sitosterol and campesterol in major quantities.

Regarding vitamins (fat and water soluble), peach palm fruit is seen as a good source of vitamins A, B, and C, in addition to the mesocarp being rich in α-tocopherol, similar to other species in the Amazon region [107,114,115]. Yuyama et al. [115] evaluated the bioavailability of vitamin A from the peach palm using the preventive method in rats. They found that the peach palm possesses a high bioavailability of this vitamin, with a relative efficiency of 250.8% when compared with the respective control groups (100%).

The identification of polyphenols in peach palms is recent, and several compounds belonging to different classes have been identified. Simple phenolic compounds, such as protocatechuic acid phenylpropanoids, were identified in the mesocarp [110]. Trace amounts of the flavonoids apigenin and myricetin, in addition to flavonoid di-*C*-glycosyl flavones, were identified in the cooked pulps of yellow and orange varieties. Among these, schaftoside (Figure 3) was the major compound in both, while vicenin-2 was detected at higher concentrations (21% of total phenolics) in the pulp of orange peach palm fruits, contrasting with the 6% found in yellow fruits. Other identified compounds were vitexin, isovitexin, isovitexin sulfate, vicenin-1, vicenin-3, isoschaftoside, and neoschaftoside [113]. Phenolic compounds act as antioxidants, and the long-term use of these compounds is encouraged for the prevention of cardiovascular and degenerative diseases [116].

The aroma compounds of the fresh fruit pulps were recently described by Faria et al. [110]. Different classes of compounds were detected, with esters (methyl salicylate) and alcohols (1-hexanol) as main components (Figure 3). Peach palm aroma composition has a contrasting profile when compared to other Amazonian fruits, which were dominated by terpenes, mainly mono- and sesquiterpenes. The consumption of this product is endorsed by its composition and high benefits to human health.

Despite the current knowledge regarding the chemical composition of fruit tissues, further efforts on metabolome analyses and the visualization of large-scale metabolite profile data should be performed and associated with metabolic pathways. This procedure, coupled with new tools for metabolite analyses, may provide useful information about the global biochemical regulation of fruit ripening. Understanding the biosynthetic pathways and their regulation is essential for the crucial biotechnological significance of the commercial production of specialized metabolites in fruit tissues that might be produced or explored in this to-date neglected peach palm fruit tissues.

### 4.3. Beside the Fruit, Peach Palm Agro-Industrial Residues Has Been Focus of Research

It is speculated that peach palms attracted the attention of Native Americans by the stem mechanical properties used for manufacturing house decorations, musical instruments, and tools as arrows and bows for hunting [2,10,74]. This is related to the high lignocellulosic material found in peach palm wood. The peach palm stem’s flexural property varies from the hardness of *maçaranduba* wood which has the flexibility [117]. Another property of peach palm wood is abrasion resistance, which allows the material to be manipulated without loss of quality, and resistance to water-based composts [118].

The increased interest in replacing synthetic fibers in favor of agro-industrial residue exploitation, aiming at cheaper raw materials with diverse applications, encourages the investigation of the peach palm’s potential as a supplier of natural fibers [75,119]. The haustorium residues can be used for crafting agglomerate panels for interior use. The best results in “finishing” and durability are seen when these fibers are combined with coir (*Cocos nucifera*) [120]. In addition, cellulose nanofibrils can also be used as the basis for biodegradable packaging or for stable emulsions, which can be applied in medicinal and cosmetic industries [121,122]. Agro-industrial waste can also be used as substrate metabolites prospection. Some examples are the association with *Ganoderma lucidum* for the production of a biosorbent, used to remove dyes present in wastewater from the textile industry [123], and as a substrate for xylanase production (for the food industry) when fermented using *Trichoderma stromaticum* [124], or for obtaining xylooligosaccharides with a higher antioxidant capacity [125]. These initiatives present a new perspective on this species’ exploitation and support the valorization of peach palm waste.

## 5. Genetic Resources: Conservation, Breeding and In Vitro Culture

### 5.1. Breeding and Implications for Genetic Erosion and Conservation

The peach palm gained international attention due to its potential as food insurance for a hungry world and as an economic resource, aspects which were highlighted in 1970 by Camacho and Soria [8,44]. Studies with *B. gasipaes* started in the same decade with the acquisition of foreign seeds, the first results of which were published in 1978 by the Agronomic Institute of Campinas (IAC), National Institute for Amazonian Research (INPA), and The Executive Commission for Cocoa Cultivation Planning (CEPLAC). The first main area of peach palm plantations (for extraction of palm hearts) was in association with the BONAL farm (Rio Branco, Acre) [44].

With the establishment of an ideotype for palm heart and fruit production, the modern genetic improvement of peach palm started in the 80s with the establishment of germplasm banks, focusing the hybridization and production of high-quality seeds. However, this model of selection is considered difficult since the peach palm is a perennial crop and the selected characters have low heritability [126]. In addition, due to institutional stagnation and germplasm banks high-costs manutention, over time, many breeding programs have been paralyzed or discontinued [1,127].

On the other hand, landrace populations that fit into the ideotype’s physical characteristics were found; the Pampa Hermosa landrace, from Yurimaguas (Peru), with between 60 and 80% of spineless plants; the Guatuso landrace from San Carlos (Costa Rica), with between 15 and 30% of spineless plants; the Putumayo genotype from Benjamin Constant (Brazil) with between 15 and 25% of spineless plants [44,126,127]. With the decrease in the natural populations of *Euterpe* spp. and lack of enhanced seeds, the producers sought the spineless seeds by themselves, introducing germplasm with no phytosanitary and quality controls [44].

Genetic manipulation implies two main factors: the conservation of germplasm and genetic erosion. Cornelius et al. [6] commented that the intensive selection of a breeding population and the use of seeds from limited germplasm are ways in which breeding programs could lead to inbreeding depression and a loss of genetic diversity. With the discovery of spineless populations, a good part of the plantations was established from these seeds. In addition to no care for genetic management, this activity puts the cultivated genotypes at risk of genetic erosion. Furthermore, wild populations are also suffering from genetic erosion due to the expansion of the Amazonian arc of fire. In some populations, it is no longer possible to guarantee the seed bank, which can lead to the extinction of the population [7,48].

High-density germplasm banks with wild and cultivated genotypes were established to support the early genetic programs. The atual active germplasm banks are Embrapa Acre; Embrapa Amapá, INPA and Embrapa Oriental (Brazil); Coorpica (Colombia); INIAP (Ecuador); INIA/ICRAF (Peru) and INIA (Venezuela) [128]. The success of genetic enhancement programs is associated with some factors; the complete characterization of germplasm banks, prospection of landraces, characterization of products chemical composition, adaptation of propagation methodologies, determination of physiologic parameters, and enlargement of phytosanitary research are some examples [126]. Today, most of remaining germplasm banks have not been fully characterized, and this creates a huge delay in prospecting selected progenies in comparison to the date of creation in these programs. Even with the selection of some high-quality progenies for both fruit and palm heart, the production of improved seeds continues to be small [48].

Germplasm banks play an important role in germplasm conservation for the genetic transformation of food crops [129]. However, associating different germplasms with on farm or in situ conservation seems to be an alternative to the coexistence of conservation and genetic improvement and for a reduction in maintenance costs [6,130]. A small in situ conservation project was successfully established at the Brazilian Agricultural Research Corporation (EMBRAPA/Acre) and associated with the RECA project in 1997, which is still responsible for agroforestry systems that focus on the recuperation of degraded areas, the conservation of biodiversity, and sustainable development of local communities. The peach palm germplasm conserved on RECA’s land is from Benjamin Constant spineless plants, and the matrix selection is based on this population with attention to spineless seeds and the commerce of the palm hearts [126]. Additionally, the World Agroforestry Centre (ICRAF) and the Peruvian National Institute for Agricultural Research and Extension (INIEA) developed a participatory improvement program in Peru (1997). With a focus on agroforestry systems and smallholders’ necessities, fruit and palm heart production, and the development of local markets were stimulated [6].

In this context, with previous efforts in germplasm banks establishment, Clement et al. [1] endorsed the idea that the expansion of peach palm germplasm banks in the tropics is not necessary. Working with genotypes of the current germplasm banks and with the molecular analyses of wild varieties, whether associated or not with agroecological systems seems, this seemed a more practical and sure way of the genetic enhancement of peach palms.

### 5.2. In Vitro Culture as a Perspective for Advances in Breeding

Different protocols for the in vitro propagation of peach palms have already been developed. In vitro cultures consist of the multiplication of cells, tissues, and organs in a controlled environment to obtain pathogen-free plants and assist in genetic enhancement [131]. Morphogenesis occurred through organogenesis, which is considered to be a natural regenerative strategy of plants and relies on the pluripotent acquisition of somatic cells, thus regenerating the organ or the plant (de novo organogenesis) [132], or by somatic embryogenesis (SE), which consists of the recovery of the cells’ totipotency through dedifferentiation, without a direct vascular connection with the explant [133,134]. Both morphogenic pathways can occur in the direct or indirect (through callogenesis) regeneration induced by growth regulators.

Indirect organogenesis was first documented by Arias and Huete [135], who pointed out the stimulatory effect of 2,4-dichlorophenoxyacetic acid (auxin analog) in callus formation. The same growth regulator was used by Stein and Stephens [136] for SE induction. Valverde et al. [137] demonstrated the influence of picloram (also an auxin analog) in callogenesis and SE induction. Almeida and Kerbauy [138] documented the direct organogenesis with the use of flower buds. However, these protocols often had random results when replicated and low rates of plantlet conversion or regeneration. A high level of explant oxidation was also observed, demonstrating the need for protocol optimization.

Steinmacher et al. [139,140,141] developed protocols for SE induction through zygotic embryo cultures, immature inflorescences, and the use of techniques known as a thin cell layer (TCL) [142]. This consists of using small slices of meristematic tissue from basal, medial, and apical plantlet meristem. These results reaffirmed the positive interaction between picloram and different peach palm explants for SE induction and also validated the applicability of indirect SE for this plant.

Steinmacher et al. [4] mentioned that the use of the temporary immersion system and cyclic cultures could shorten and scale in vitro multiplication. The use of the RITA system helps to enhance the embryogenic capacity, protein, and starch content, lower the DNA global methylation rate, and lower alcohol dehydrogenase activity [143]. Padilha et al. [144] also applied a temporary immersion system (TIS) to promote plantlet rooting, creating a successfully improved protocol for plantlet conversion.

In a recent study, Campos-Boza et al. [145] applied protocols using shoots while also testing different concentrations of picloram and different positions of TCL meristems in the culture medium. The study concluded that there were no significant differences in the picloram concentrations and meristem position in embryonic competence acquisition, contrasting observations by Steinmacher et al. [141]. These results may be related to the phenological phase of the tissue. Other experiments used plant material from a juvenile or immature phase (seedlings, immature inflorescences, zygotic embryos), unlike the shoots used by Campos-Boza et al. [145]. The material provided from shoots showed less contamination and browning in comparison with cryopreserved zygotic embryos and TCLs from seedlings. The induction phase was precocious (<70 days), and the conversion phase generated 12 plantlets with additional lateral shoots and roots, with a rate of 60% successful conversion. Although the plantlet yield was low, this study attested shoot viability as vegetative material for adult peach palm propagation.

Therefore, it is possible to state that the in vitro cultivation of peach palms, aiming at the regeneration of seedlings by somatic embryogenesis, is possible. It is important to point out that further studies should seek to reduce the level of contamination of explants, as well as to increase the number of embryogenic calluses and the conversion rate, in order to create more effective protocols. In addition, the application of immersion systems is seen as an alternative due to the increase in seedling production in less time. Finally, advances in the in vitro cultivation of this species open space for the genetic improvement of the species, as well as being an alternative for the conservation of threatened genotypes.

## 6. Conclusions and Future Directions

Centuries of domestication have provided humanity with an immense variety of peach palm genotypes whose economic and production potential has not yet been fully exploited. In addition to the relative knowledge of peach palm cultivation traits, the lack of enhanced seeds and inefficiency of macropropagation configure a barrier to peach palm cultivation expansion.

Still, the most profitable activities are palm heart and fruit commercialization. Peach palm fruits present themselves as nutritious food, with a good quantity of specialized metabolites that benefit human health but are also desired by the pharmaceutical and food industries. In addition to direct consumption, the fruit can also provide by-products of high nutritional value. Finally, not only does the fruit have economic potential, but also other possibilities, such as agro-industrial residues, which are considered a cheap source of natural fibers. The manufacturing of fiber panels, the obtention of cellulose nanofibrils for biofilms reinforcement, and the use of lignocellulose as a substrate for the prospection of prebiotics and other molecules are good examples of peach palm agro-industrial waste biotechnological applications.

The lack of complete molecular characterization of germplasm banks associated with the inexistence of an efficient in vitro protocol was pointed out as the main limitation in peach palm genetic transformation. These barriers have been surpassed in the last few years, providing a clear view of the landraces diversity that is available for conservation and genetic enhancement. However, genomic information and transcriptome analyses are still limited to peach palm fruits. The production of such a large amount of nucleotide sequence data using next-generation technologies is fundamental for biotechnological applications of distinct aspects of *B. gasipaes*. Thus, with the recent plastome sequencing of both varieties of *B. gasipaes*, advances in the establishment of efficient in vitro cultivation protocol, and better recognition of the plant’s exploitable biological aspects, means that doors for genetic engineering with peach palms are opening.

In summary, here we present an illustration of the combined use of the current knowledge (technical-scientific) about the species and genetic tools coupled with adaptive fitness-related quantitative markers to enhance our understanding of underutilized tropical species in the framework of a biotechnological program involving *Bactris gasipaes*. Further studies investigating whether epigenomic and phenotypic variations are present in tropical species may explain their responses to environmental stress conditions. Indeed, *B. gasipaes* presents itself as a useful non-model species for understanding the biotechnological significance (or lack thereof) of neglected and underutilized plants with virtually no genetic improvement in a changing climate.

## Figures and Tables

**Figure 1 plants-12-00337-f001:**
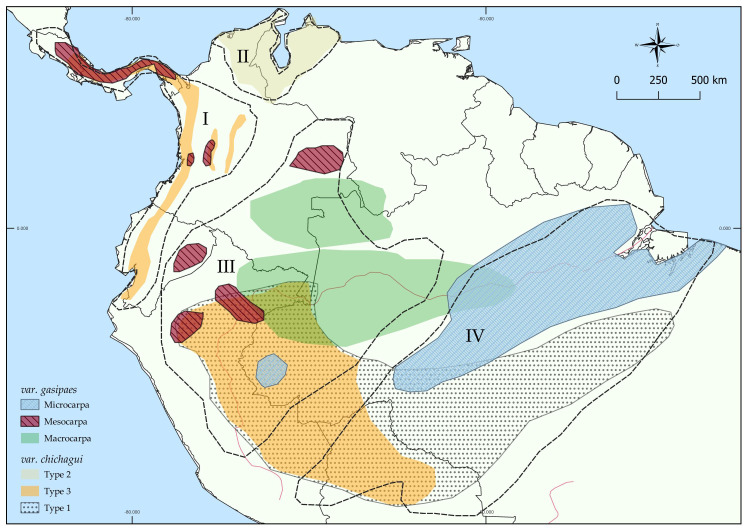
Distribution of *B. gasipaes* through Central and South America (adapted from Clement et al. [18] and Hernández-Ugalde et al. [20]). The dashed lines comprehend the four main complexes: I. Occidental group: composed of representants of var. *chichagui* (Upala, Azuero, Daríen, Chontila, and Chinamato) and var. *gasipaes* (Utilis landrace); II. Maracaibo group: comprehending *B. caribaea* and *B. macana* (outgroups); III. Upper Amazonia: composed of Caquetá, Putumayo (Yurimaguas population), Vaupés, Inirida, Ayacucho, Pastaza, Tigre, and Juruá landraces; IV. Eastern Amazonia: Pará and Tembé landraces, and var. *chichagui* Acre and Xingú.

**Figure 2 plants-12-00337-f002:**
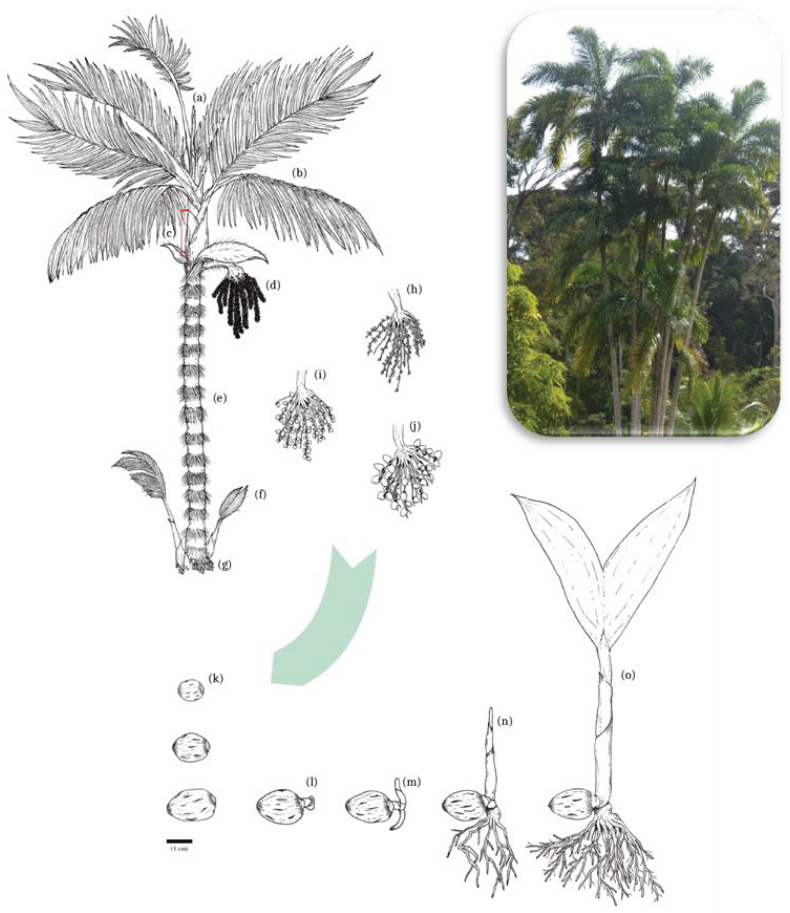
General morphology of *B. gasipaes* and germination. On the right, a Peach palm near to an ombrophilous dense forest in Altamira, Para, Brazil, followed by a representation of an adult individual and different fruits sizes. The arrow indicates different seed sizes and the germination morphology, adapted from Silva et al. [24]. Legends: (a) apical gem; (b) pinnate leaves; (c) palm heart; (d) inflorescences; (e) internode with spines; (f) shoots; (g) roots; (h–j) different sizes of fruits; (k) seeds; (l) non-differentiated cell mass; (m) radicular and caulinar primordium; (n) eophyll elongation; (o) bifid primary leaf.

**Figure 3 plants-12-00337-f003:**
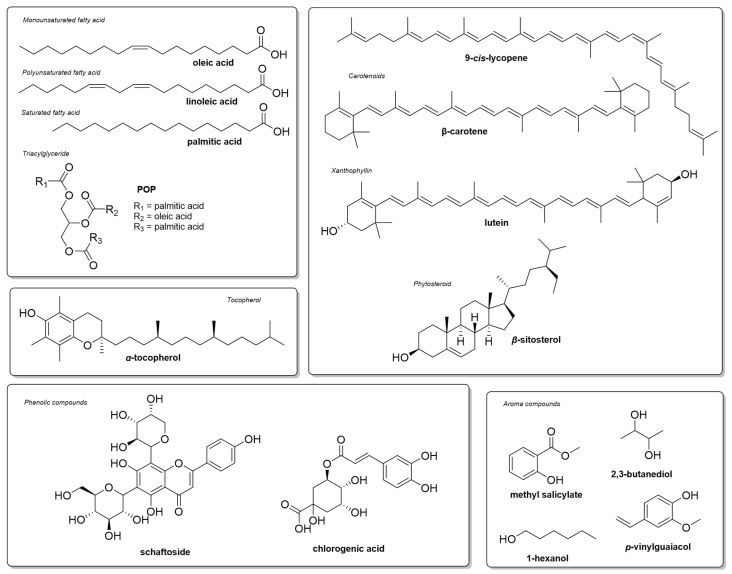
Chemical structures of the major metabolites described for the peach palm.

**Table 1 plants-12-00337-t001:** Nutritional aspects of peach palm fruit pulp.

	*Bactris gasipaes*	*Mauritia flexuosa*	References
Moisture (%)	45–65.1	50.5–79.35	[90,95,96,97]
Starch (%)	27–59.5	7.28–36.3	[89,95,98]
Protein (%)	2.12–14.7	1.8–3.7	[89,92,95,99]
Oils (%)	5.57–27	11.20–19.0	[89,95,96,99]
Total fiber (%)	1.25–6.6	7.9–22.8	[90,92,95,99]
Ashes (%)	0.6–0.9	0.6–0.8	
Energy (Kcal 100 g^−1^)	351.4	189.6–1006	
Minerals (100 g)			
K (mg)	206.4–289.3	183.55–919.6	[90,93,97,100]
Ca (mg)	10.2–24.7	35.4–132	
Mg (mg)	16.9–17.6	14.29–60.2	
Na (mg)	0.2–12.6	134.4	[90,93,100]
Fe (mg)	0.47–0.74	0.69–4	[90,97,100]
Cu (mg)	-	0.61	[100]
Zn (mg)	0.26–0.28	1.08	[90,93,100]
Mn (mg)	0.08–0.11	8.72	
Cl (µg)	7.6–30.7	-	[90]
Cr (µg)	8.2–13.9	-	
Se (µg)	3.3–11.4	-	
Rb (µg)	491.4–924.1	-	
Br (µg)	34.3–189.4	-	
Ba (µg)	103.9–164.5	-	
Pa (µg)	56.4–60.9	-	
Ce (µg)	1.3–2.1	-	
La (ng)	70.5–521.8	-	
Sb (ng)	31.0–99.0	-	
Au (ng)	30.3–57.8	-	
Sc (ng)	7–10.4	-	
Fatty acids (%)			
Palmitic (16:0)	24.1–39.6	18.9	[90,99]
Palmitoleic (16:1)	5.2–7.4	0.3	
Margaric (17:0)	-	-	
Stearic (18:0)	0.8–1.7	1.3	
Oleic (18:1)	42.8–60.8	75.7	
Linoleic (18:2)	1.2–1.4	2.1	
Linolenic (18:3)	0–1.8	-	
Arachidonic (20:0)	-	1.7	
Amino acids			
	*Bactris gasipaes*	FAO	
Essential (mg g^−1^)			
Histidine	0.09	16	[90,101,102]
Isoleucine	0.16–1.70	13	
Leucine	0.28–3.14	19	
Lysine	0.21–1.67	16	
Methionine	0.08–0.8	17 ^a^	
Phenylalanine	0.14–2.04	19 ^b^	
Threonine	0.18–2.71	9	
Valine	0.19–2.83	13	
Tryptophan	0.45	5	
Non-essential (ug g^−1^)			
Alanine	3.51	-	[90,101]
Arginine	0.29	-	
Aspartate	4.33	-	
Serine	2.72	-	
Glutamate	4.98	-	
Glycine	0.27–2.87	-	
Tyrosine	0.14	-	
Proline	2.57	-	

Associated with cysteine ^a^ or tyrosine ^b^.

**Table 2 plants-12-00337-t002:** Specialized compounds in peach palm pulp.

Terpenoids (μg/g)	*Bactris gasipaes*	*Mauritia flexuosa*	References
*cis*-γ-Carotene 1	3.2	-	[106]
*cis*-γ-Carotene 2	2.3	2.33	
*cis*-γ-Carotene 3	2.1	9.88	
*cis*-γ-Carotene 4	28.3	-	
*cis*-γ-Carotene 5	0.13	-	
*cis*-δ-Carotene 1	5.2	5.46	
*cis*-δ-Carotene 2	2.1	3.67	
*cis*-δ-Carotene 3	0.86	2.42	
*cis*-β-Zeacarotene 1	-	-	
*cis*-β-Zeacarotene 2	-	-	
*cis*-Violaxanthin	-	-	
*cis*-Neoxanthin	-	-	
*cis*-Lutein	-	-	
9-*cis*-Lycopene	8.4	-	
9-*cis*-β-Carotene	2.2	18.57	
13-*cis*-β-Carotene	4.02	59.23	
15-*cis*-β-Carotene	0.08	8.87	
*all-trans*-α-Carotene	1.8	3.23	
*all-trans*-α-Cryptoxanthin	0.12	1.28	
*all-trans*-β-Carotene	55.5	372.32	
*all-trans*-β-Cryptoxanthin	-	-	
*all-trans*-β-Zeacarotene	-	-	
*all-trans*-δ-Carotene	45.8	2.09	
*all-trans*-γ-Carotene	35.4	14.76	
*all-trans*-ζ-Carotene	-	0.08	
*all-trans*-Neoxanthin	-	-	
*all-trans*-Zeaxanthin	-	-	
5,6-epoxy-β-Carotene	-	0.41	
5,6-epoxy-β-Cryptoxanthin	-	0.1	
5,8-epoxy-β-carotene	0.03	7.44	
Phytoene	-	0.34	
Zeaxanthin	-	-	
*all-trans*-Lutein	-	0.03	
di-*cis*-α-Carotene	-	1.25	
Vitamins (100 g)			
Thiamine (μg)	-	-	[92]
Riboflavin (μg)	-	-	
Niacin (mg)	0.13	-	
Ascorbic acid (mg)	0.9–14	13	[92,107]
α-Tocopherol (mg)	11.7	110–197	[108,109]
β + γ-Tocopherol (mg)	-	476	
δ-Tocopherol (mg)	-	44.1	
Phenolic compounds (μg g^−1^)			
Apigenin	0.002	102.48	[110,111]
Caffeic acid	-	895.53	
Chlorogenic acid	0.02	1154.15	
Ferulic acid	0,16	184.66	
Kaempferol	-	41.54	
Luteolin	-	1060.9	
Myricetin	0.02	145.11	
Protocatechuic acid	0.03	2175.93	
*p*-Coumaric acid	0.01	277.74	
Quercetin	-	83.27	
Quinic acid (mg g^−1^)	-	230.71	
(+)-Catechin	-	961.21	
(−)-Epicatechin	-	1109.93	
Phytosterols (mg 100 g^−1^)			
β-Sitosterol	8.22	7.66	[108]
Campesterol	1.09	1.39	
Stigmasterol	0.42	0.81	
Δ^5^-Avenasterol	0.27	0.14	

## Data Availability

Not applicable.

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
