# Peer review of "Understanding the Technical-Scientific Gaps of Underutilized Tropical Species: The Case of Bactris gasipaes Kunth"

_plants, 2023, doi:10.3390/plants12020337_

Round 1
Reviewer 1 Report
The structure of the manuscript is logical and provides a comprehensive overview of the use and possibilities of the Bactris gasipaes palm species. The authors have processed a fairly large amount of literature, 130 sources.
Why do the authors use the term "biotechnology"? Biotechnology means the application of biology in technology, and today it primarily means the manipulation of genes. It is not enough if the authors say in the summary (line 599/600) "/---/ or as a to produce other molecules of biotechnological interest." Such a sentence can be said about every natural species, that this particular species can be of interest in biotechnology - all plants, animals, fungi have virtues that can be of interest to biotechnology. Therefore, if the authors do not point out a specific characteristic that other species do not have, and which exists only in this species, then such a conclusion is misleading. If the authors find properties in this palm that other species do not have, they can only say that it can be of interest in the development of biotechnology.
The authors only mention "neglected" in the title and summary, but do not mention in the text how this species has been neglected? If this species has been used for centuries and is still being used, it cannot be said that it is "neglected".
Since you have also shown the contours of the country borders on Figure 1, I recommend that you include the names of the countries where this palm tree is found in the introduction. E.g. "This palm is native to the in America in Brazil, XXXX, XXXX, and cultivated in Brazil, XXXX, XXXX."
Line 34. Plant names that are not in English should be put in italics and it would be good to note in which languages this palm tree are called so?
Line 66/69. If it is known, you should also list which "Native Americans" peoples we are talking about? And what "weapons" did the "Native Americans" make of this palm?
You say in Figure 4 that "Vermicide" is obtained from palm roots, but there is no reference to this in the text? You also point in figure out that "Inflorescences" is used for "Condment", this fact is also not mentioned in the text. Where did you get that fact?
Currently, one term is given in several different ways in the text: (singular) “heart of palm”, “palm-heart”, “palm heart”; (plural) “hearts of palm”, “palm-hearts”, “hearts-of-palm”. Make this term uniform throughout the text.
Author Response
Reviewer 1
The structure of the manuscript is logical and provides a comprehensive overview of the use and possibilities of the Bactris gasipaes palm species. The authors have processed a fairly large amount of literature, 130 sources.
Author's Response: Thank you very much. We are very happy to receive the positive comments made by the reviewer on our article.
Why do the authors use the term "biotechnology"? Biotechnology means the application of biology in technology, and today it primarily means the manipulation of genes. It is not enough if the authors say in the summary (line 599/600) "/---/ or as a to produce other molecules of biotechnological interest." Such a sentence can be said about every natural species, that this particular species can be of interest in biotechnology - all plants, animals, fungi have virtues that can be of interest to biotechnology. Therefore, if the authors do not point out a specific characteristic that other species do not have, and which exists only in this species, then such a conclusion is misleading. If the authors find properties in this palm that other species do not have, they can only say that it can be of interest in the development of biotechnology.
Author's Response: We agree with your comment that biotechnology is the use of biology to develop new products, methods, and organisms with a focus on improving the environment, agriculture, industry, health, and society. Notably, biotechnology has existed since the beginning of civilization with the domestication of plants, microorganisms, and animals. Within our manuscript, we intentionally want to promote peach palm as a fruit crop it is important to understand the current state-of-art of the species. Therefore, we focused on updating technical knowledge, existing limitations, and advances in peach palm with an emphasis on the potential biotechnological applications of this rather important yet neglected specie. By doing that, we finally discuss how our current knowledge may be integrated, with the aim of identifying profitable activities involved in the peach palm market and discovering novel products with biotechnological applications. Please see the updated manuscript for further details of what has been modified accordingly. We hope this comes across more clearly in the updated new version.
The authors only mention "neglected" in the title and summary, but do not mention in the text how this species has been neglected? If this species has been used for centuries and is still being used, it cannot be said that it is "neglected".
Author's Response: Although we can clearly understand the reviewer's criticism, we do not fully agree with it. First, in this updated new version we have mentioned the term “neglected” a few more times and secondly, although this species has been used for centuries, its domestication has been hampered due to distinct challenges (discussed in detail within the manuscript) and its potential biotechnological applications were not developed due to a lack of studies. In summary, we posit that this is an underutilized and neglected specie since it harbors tremendous potential and has not been properly used to date. We further believe that this review will open up a new avenue of research using this specie not only in Brazil but also in the other several countries where it is usually cultivated with economic importance. Please see the updated manuscript for further details of what has been modified accordingly. We hope this comes across more clearly in the updated new version.
Since you have also shown the contours of the country borders on Figure 1, I recommend that you include the names of the countries where this palm tree is found in the introduction. E.g. "This palm is native to the in America in Brazil, XXXX, XXXX, and cultivated in Brazil, XXXX, XXXX."
Author's Response: Thank you very much for your careful and thorough revision of our manuscript. It has been modified accordingly.
Line 34. Plant names that are not in English should be put in italics and it would be good to note in which languages this palm tree are called so?
Author's Response: Thank you very much. We have modified it accordingly by adding not the languages but rather the country where it is usually denominated.
Line 66/69. If it is known, you should also list which "Native Americans" peoples we are talking about? And what "weapons" did the "Native Americans" make of this palm?
Author's Response: Thank you very much for your careful and thorough revision of our manuscript. It is unclear which native Americans are since we are quoting others’ manuscripts. To avoid any misinterpretation, we have removed the word “weapons” since it cannot be properly defined. We hope this comes across more clearly in the updated new version.
You say in Figure 4 that "Vermicide" is obtained from palm roots, but there is no reference to this in the text? You also point in figure out that "Inflorescences" is used for "Condment", this fact is also not mentioned in the text. Where did you get that fact?
Author's Response.
Currently, one term is given in several different ways in the text: (singular) “heart of palm”, “palm-heart”, “palm heart”; (plural) “hearts of palm”, “palm-hearts”, “hearts-of-palm”. Make this term uniform throughout the text.
Author's Response: Thank you very much. It has been modified accordingly and we have used throughout the manuscript the term “peach palm”. We hope this comes across more clearly in the updated new version.

Reviewer 2 Report
I enjoyed reading the manuscript as the review helps me to update the knowledge about peach palm nicely. However, I also saw some weakness of the paper, largely for logical components. Among those I would concern more are:
1. The review does not give us the complete picture why peach palm is a neglected tropic species. For example, information on current production, economic component of it. It would be nice to show all the barriers known so far so that the species is not well explored.
2. The term of "biotechological potential" is not defined and really vague, so this would affect the message deliverable of this paper. For example, I read the abstract and the conclusion section, and I cannot figure out what are the major message for biotechnology here. Actually, if I were the authors, I would have a simple title like: How little is known about peach palm, for the content as it is.
3. I also made some editorial changes and comments directly on the pdf file attached for the authors to consider.

Author Response
Reviewer 2
I enjoyed reading the manuscript as the review helps me to update the knowledge about peach palm nicely. However, I also saw some weakness of the paper, largely for logical components. Among those I would concern more are:
Author's Response: Thank you very much. We are very happy to receive the positive comments made by the reviewer on our article.
- The review does not give us the complete picture why peach palm is a neglected tropic species. For example, information on current production, economic component of it. It would be nice to show all the barriers known so far so that the species is not well explored.
Author's Response: Thank you very much. We are very happy to receive such positive comments. Following the reviewer's comment, we have modified the Introduction. First, in this updated new version we have mentioned the term “neglected” a few more times and secondly, although this species has been used for centuries, its domestication has been hampered due to distinct challenges (discussed in detail within the manuscript) and its potential biotechnological applications were not developed due to a lack of studies. In summary, we posit that this is an underutilized and neglected specie since it harbors tremendous potential and has not been properly used to date. We further believe that this review will open up a new avenue of research using this specie not only in Brazil but also in the other several countries where it is usually cultivated with economic importance. Please see the updated manuscript for further details of what has been modified accordingly. We hope this comes across more clearly in the updated new version.
- The term of "biotechnological potential" is not defined and really vague, so this would affect the message deliverable of this paper. For example, I read the abstract and the conclusion section, and I cannot figure out what are the major message for biotechnology here. Actually, if I were the authors, I would have a simple title like: How little is known about peach palm, for the content as it is.
Author's Response: Thank you very much. Following the reviewer's comment, we have modified significant parts of our manuscript in an attempt to properly describe the term “biotechnological potential”. We further attempted to make it clear what we intended to cover and such changes can be seen in the updated manuscript for further details. Once again, thank you very much for the highly valid comments.
- I also made some editorial changes and comments directly on the pdf file attached for the authors to consider.
Author's Response: We thank the reviewer for his/her suggestions and corrections and apologize for such mistakes. We have done all the modifications suggested and taken all of them into account in producing the attached new version. It is needless to mention that all comments were very valid and helped us to improve the new version of the manuscript. We truly thank the reviewer for the thorough revision of our work.
To summarize, we have addressed all comments and have corrected all minor and major mistakes outlined by the reviewers included also in the scanned version. The revised portions are highlighted in yellow and written in red in the text and the main revisions corresponding to the reviewers’ comments are presented in a detailed point-by-point in the annotated summary listed below (see marked track changes file for further details).
The manuscript has not been submitted for publication elsewhere. Accordingly, I would like to state, on behalf of all authors, that they have read and contributed to the work described here and take entire responsibility for it. There is also no conflict of interest. We are confident that the new updated version of our manuscript is highly improved and meets the high standard required for publication in Plants.

Round 2
Reviewer 2 Report
It seemed that the authors have considered my comments into the revision. Even though with the addition of Section 6, however, I don't think the revision has the clear messages and the title is still misleading. It seemed that the authors did not understand my comments, along with those from Reviewer#1, on these points. For example, biotechnology potential is not the same as biotechnology application potential or potential biotechnology application. As I read through the revision, I did not get any of three meanings either. Please read over your abstract and your conclusion, you would see if your title is not associated with those points in abstract and your conclusion. The authors did not like my previously suggested title, and here I showed another one: Exploration of underutilized tropical species: a case xxx.
More I found many English issues in the places where new revision was made and highlighted in yellow. One example is enough; see lines 640-644. To me, you don't need to define what is neglected or underutilized, but you need to know and show the issues or features this species is not fully utilized yet, such as the production is so slow (in acre), etc. However, reading the revision, I think it has been widely utilized across the south America, but many potentials are not fully explored to generate a full value of this species yet. Now you may know partly why your title is misleading.
In short, you need to re-configure your messages and title, explaining why I decide "reconsider after MAJOR revision. Good luck,.
Author Response
Responses to the reviewer’s comments
To: Dr. Dilantha Fernando
Editor-in-Chief
Plants
Cc: Ms. Jia Yi Huin
Assistant Editor
Plants
E-mail: jiayi.huin@mdpi.com
Ref. No.: plants-2114578_R2
Title: Biotechnological potential of neglected tropical species: the case of Bactris gasipaes Kunth
Dear editors,
Please, find enclosed the revised version of our manuscript entitled “Biotechnological potential of neglected tropical species: the case of Bactris gasipaes Kunth (manuscript ID plants-2114578_R2). We appreciate the comments and the suggestions/corrections made by the reviewer. We believe that the suggestions contributed significantly to the improvement of our manuscript. Overall, we accepted almost all comments from the reviewers and from you. The changes were highlighted in red here (author’s responses) and also in the manuscript (track change version). You may find the amendments to our revised manuscript and responses to the reviewers' comments in the changes below:
Comments from the reviewer and author's responses:
Reviewer# 2: It seemed that the authors have considered my comments into the revision. Even though with the addition of Section 6, however, I don't think the revision has the clear messages and the title is still misleading. It seemed that the authors did not understand my comments, along with those from Reviewer#1, on these points. For example, biotechnology potential is not the same as biotechnology application potential or potential biotechnology application. As I read through the revision, I did not get any of three meanings either. Please read over your abstract and your conclusion, you would see if your title is not associated with those points in abstract and your conclusion. The authors did not like my previously suggested title, and here I showed another one: Exploration of underutilized tropical species: a case xxx.
Authors' responses: Our sincere thanks go out to you for considering that a least some of your comments were satisfactorily addressed. In this second review, we focused on the questions that have yet to be fully answered. Here, to better describe what we changed, we highlighted the title alteration, which has been changed from Biotechnological potential of neglected tropical species: the case of Bactris gasipaes Kunth to Understanding the technical-scientific gaps of underutilized tropical species: the case of Bactris gasipaes Kunth. We checked the alignment between the title, abstract, and conclusions. In addition, the keywords were also changed. In this version, we have also included the graphical abstract. In short, given the arguments set out above, we look forward to receiving your positive appreciation of what we improved in the second version of our manuscript.
Reviewer# 2: More I found many English issues in the places where new revision was made and highlighted in yellow. One example is enough; see lines 640-644. To me, you don't need to define what is neglected or underutilized, but you need to know and show the issues or features this species is not fully utilized yet, such as the production is so slow (in acre), etc. However, reading the revision, I think it has been widely utilized across the south America, but many potentials are not fully explored to generate a full value of this species yet. Now you may know partly why your title is misleading.
Authors' responses: Thank you for your appreciation and all the helpful criticism. Yes, we agree with the general comments made by the reviewer! We promptly accepted and checked the whole manuscript, so we tried to improve the English language. However, we would like a few more days to resent our manuscript and submit a more polished version. We would appreciate an extension of one week. Would that be possible? Please let us know!
Reviewer# 2: In short, you need to re-configure your messages and title, explaining why I decide "reconsider after MAJOR revision. Good luck.
Authors' responses: We are pleased to receive such comments, and we truly appreciate the contribution´s reviewer#2 to our manuscript. In brief, all the comments made by his/her were promptly accepted by the authors. Without a doubt, your suggestions certainly significantly contributed to improving the overall quality and integrity of our review article. To meet the reviewer's suggestions, our manuscript was re-configured to reposition the topics hierarchically arranged by subject - from general to specific. In other words, the title of the manuscript has been rewritten, while the abstract has been reformulated, following an alignment through of text until the conclusions.
Finally, thank you once again for the opportunity to improve our study, if you have any questions, please let us know and we shall be happy to include them.
Sincerely,
Sincerely,
Wagner and José Francisco (on behalf of the co-authors)
Prof. Wagner L. Araújo
Departamento de Biologia Vegetal
Universidade Federal de Viçosa
36570-900 Viçosa, MG, Brazil
Tel: +55 (31) 3612.5358
E-mail: wlaraujo@ufv.br
And
José Francisco de Carvalho Gonçalves, Ph.D.
Senior Scientific Researcher
National Institute for Amazonian Research (INPA-MCTI)
Laboratory of Plant Physiology and Biochemistry / PO Box 2223
Lattes ID: http://lattes.cnpq.br/0553096006639259
Orcid ID: https://orcid.org/0000-0001-9197-4617
Phone: 55 92 36431938
Manaus-AM, Brazil
